# Classifying Unconscious, Psychedelic, and Neuropsychiatric Brain States with Functional Connectivity, Graph Theory, and Cortical Gradient Analysis

**DOI:** 10.3390/brainsci14090880

**Published:** 2024-08-30

**Authors:** Hyunwoo Jang, Rui Dai, George A. Mashour, Anthony G. Hudetz, Zirui Huang

**Affiliations:** 1Neuroscience Graduate Program, University of Michigan, Ann Arbor, MI 48109, USA; janghw@umich.edu (H.J.); gmashour@med.umich.edu (G.A.M.); ahudetz@med.umich.edu (A.G.H.); 2Center for Consciousness Science, University of Michigan Medical School, Ann Arbor, MI 48109, USA; rudai@med.umich.edu; 3Department of Anesthesiology, University of Michigan Medical School, Ann Arbor, MI 48109, USA; 4Michigan Psychedelic Center, University of Michigan Medical School, Ann Arbor, MI 48109, USA; 5Department of Pharmacology, University of Michigan Medical School, Ann Arbor, MI 48109, USA

**Keywords:** machine learning, resting-state functional MRI, functional connectivity, graph theory, cortical gradient, unconsciousness, psychedelics, neuropsychiatric disorders, sleep, anesthesia

## Abstract

Accurate and generalizable classification of brain states is essential for understanding their neural underpinnings and improving clinical diagnostics. Traditionally, functional connectivity patterns and graph-theoretic metrics have been utilized. However, cortical gradient features, which reflect global brain organization, offer a complementary approach. We hypothesized that a machine learning model integrating these three feature sets would effectively discriminate between baseline and atypical brain states across a wide spectrum of conditions, even though the underlying neural mechanisms vary. To test this, we extracted features from brain states associated with three meta-conditions including unconsciousness (NREM2 sleep, propofol deep sedation, and propofol general anesthesia), psychedelic states induced by hallucinogens (subanesthetic ketamine, lysergic acid diethylamide, and nitrous oxide), and neuropsychiatric disorders (attention-deficit hyperactivity disorder, bipolar disorder, and schizophrenia). We used support vector machine with nested cross-validation to construct our models. The soft voting ensemble model marked the average balanced accuracy (average of specificity and sensitivity) of 79% (62–98% across all conditions), outperforming individual base models (70–76%). Notably, our models exhibited varying degrees of transferability across different datasets, with performance being dependent on the specific brain states and feature sets used. Feature importance analysis across meta-conditions suggests that the underlying neural mechanisms vary significantly, necessitating tailored approaches for accurate classification of specific brain states. This finding underscores the value of our feature-integrated ensemble models, which leverage the strengths of multiple feature types to achieve robust performance across a broader range of brain states. While our approach offers valuable insights into the neural signatures of different brain states, future work is needed to develop and validate even more generalizable models that can accurately classify brain states across a wider array of conditions.

## 1. Introduction

Brain state classification is a major goal in both clinical and research settings [1,2,3,4,5,6]. In clinical practice, precise classification and prediction of human brain states can enhance diagnostic accuracy for various neurological and psychiatric conditions, such as epilepsy [7], sleep disorders [8], and other mental health disorders [9]. This facilitates personalized and effective treatment planning [10]. Distinguishing between different brain states is also crucial for monitoring the effects of therapeutic interventions, such as anesthesia [11], neurostimulation [12], and psychotropic medications [13]. In neuroscience, brain state classification provides insights into the underlying neural mechanisms of consciousness, cognition, and behavior [14,15,16,17,18]. Identifying neural patterns associated with specific cognitive, behavioral, or clinical states can enhance our understanding of brain function and dysfunction.

Brain state classification has traditionally relied on neuroimaging techniques like functional magnetic resonance imaging (fMRI), electroencephalography, and positron emission tomography [19,20,21,22]. Among these, fMRI stands out for its high spatial resolution in non-invasive brain function analysis [23]. Resting-state fMRI offers the added advantage of studying intrinsic brain activity without task-related confounds [24].

Various machine learning algorithms such as support vector machine (SVM), random forests, and artificial neural networks have been exploited [25,26,27,28]. These computational approaches have demonstrated success in identifying reliable neural signatures that correlate with physiologically or clinically defined states, including seizure detection [7,29], sleep stage identification [30], and psychiatric condition differentiation [28].

Notably, SVM has proven to be a robust and versatile method for brain pattern recognition, as extensively reviewed in the literature [31,32,33,34]. For instance, SVM has successfully classified major depression based on fMRI signal differences observed during the emotional processing of sad faces [35]. Additionally, SVM has demonstrated over 90% accuracy in distinguishing the progression of Alzheimer’s disease by analyzing brain networks reconstructed from resting-state fMRI data [36]. Moreover, SVM has shown promising results in the classification of schizophrenia and bipolar disorder, further highlighting its utility in neuropsychiatric research [37].

One primary difficulty in brain state classification lies in subtle and overlapping neural signatures [38]. Additionally, individual variability and challenges in achieving consistent test–retest reliability of brain functions make it difficult to develop models that are universally applicable [39,40]. The influence of external factors, such as medication, environmental stimuli, and individual health conditions, further complicates the classification process. Variations in fMRI data quality and preprocessing methods such as global signal regression (GSR) can also affect the quality of classification [41]. Consequently, there is a critical need for a methodology that can be generalized to different conditions and populations.

Features that are commonly extracted for supervised learning of brain states include functional connectivity and graph theoretical measures [42,43]. Functional connectivity, often defined as the temporal correlation between blood-oxygen-level-dependent (BOLD) signals in different regions, captures the interactions between brain areas. Both regional-level and network-averaged connectivity can be used as features. Building on functional connectivity, graph theoretical measures apply network theory to brain imaging data [44]. Using principles of network science, the brain is modeled as a graph, with nodes representing brain regions and edges representing connections between them. Once reconstructed, the analysis of these graphs includes metrics such as efficiency, the clustering coefficient, modularity, and the participation coefficient. Global network efficiency reflects the ease of information exchange across the brain network, indicating global integration. In contrast, the clustering coefficient reflects the degree to which nodes tend to cluster together, indicating segregation. Modularity measures the degree to which the network can be divided into distinct communities or modules, and the participation coefficient reflects the extent to which individual nodes are connected across different modules.

An emerging and promising framework is cortical gradient analysis. Constructing functional gradients is a novel approach to understanding brain organization by spatially arranging each area on a virtual *N*-dimensional (often 1 to 3-D) continuum [45,46,47]. Unlike traditional methods that focus on discrete brain regions or networks, cortical gradients capture a holistic organization of the brain. Cortical gradients often manifest a hierarchical structure, with the major gradient axis of variation (Gradient-1) separating sensorimotor areas and higher-order association areas [46,47]. Although not yet widely used in the machine learning perspective [48,49], cortical gradients offer powerful features for brain classification by quantifying brain changes that may not be apparent when examining individual regions or networks.

This study proposes a classification methodology drawing on three complementary approaches to characterize brain function: functional connectivity measures to capture inter-regional interactions, graph-theoretical metrics to reveal network-level properties, and cortical gradients to elucidate the brain’s hierarchical organization and global functional architecture. We evaluate the performance and robustness of this approach in classifying healthy awake baseline and atypical brain states, in three typically unconscious states (NREM2 sleep, propofol sedation, and propofol general anesthesia), three psychedelic states (lysergic acid diethylamide (LSD), subanesthetic ketamine, and nitrous oxide (N_2_O)), and three neuropsychiatric states (attention-deficit hyperactivity disorder (ADHD), bipolar disorder, and schizophrenia). We train five base SVM models from three different feature types: (1) functional connectivity profiles (derived from both GSR-applied and GSR-unapplied (noGSR) BOLD signals), (2) graph metrics (from GSR and noGSR signals), and (3) cortical gradient characteristics. Three different ensemble strategies—including feature integration, hard voting (majority voting), and soft voting—are tested to leverage the strength of individual base models. Additionally, we assess cross-dataset performance to explore the model transferability. Finally, we analyze feature importance to identify the specific features that contribute most substantially to the classification accuracy (Figure 1). To this end, we present a novel brain state classification methodology that integrates functional connectivity, graph-theoretical metrics, and cortical gradients, demonstrating robust performance across unconscious, psychedelic, and neuropsychiatric states with enhanced generalizability.

## 2. Materials and Methods

### 2.1. Datasets

#### 2.1.1. Experimental Conditions

Brain state conditions utilized in this study were grouped into three meta-conditions: unconscious, psychedelic, and neuropsychiatric. The unconscious meta-condition includes N2 sleep (*n* = 29), deep sedation (*n* = 52), and general anesthesia (*n* = 23). We acknowledge that the term “unconscious” is an operational definition in this context, as consciousness exists on a spectrum [50]. N2 sleep and sedated states may retain some level of internal awareness. However, these states represent significant reductions in responsiveness and awareness compared to normal wakefulness, justifying their grouping under this label for our analytical purposes. The psychedelic meta-condition consisted of LSD (*n* = 15; two sessions per subject), subanesthetic ketamine (*n* = 12), and N_2_O (*n* = 16). The neuropsychiatric meta-condition included ADHD (*n* = 40), bipolar disorder (*n* = 49), and schizophrenia (*n* = 50). Except for neuropsychiatric conditions, we employed a within-subject design, by comparing two brain states (baseline vs. atypical) within the same participants. A detailed summary table of the datasets is summarized in Table 1.

#### 2.1.2. NREM2 Sleep

The dataset was sourced from the OpenNEURO database, provided by Pennsylvania State University, with informed consent [51,52]. It comprised 33 healthy participants experiencing wakefulness and three sleep stages (N1, N2, and N3). Sleep stages were identified using EEG signatures by a registered polysomnographic technologist. MRI scans were performed with a 3 Tesla Prisma Siemens Fit scanner, utilizing a Siemens 20-channel receive-array coil. The fMRI data were collected via an EPI sequence with a TR of 2.1 s, a TE of 25 ms, a slice thickness of 4 mm, 35 slices, a FOV of 240 mm, and an in-plane resolution of 3 mm × 3 mm. As only three participants exhibited N3 sleep, we focused on the N2 stage, present in 29 out of 33 subjects, as our target condition.

#### 2.1.3. Propofol Deep Sedation

We merged two separate datasets acquired from two previous studies conducted at the University of Michigan, both approved by the Institutional Review Board. In total, the combined dataset includes data from 52 healthy, right-handed individuals. Further details on the experimental setup are elaborated in prior publications [46,53,54].

Dataset-1 involved 26 participants who provided informed consent and were compensated for their participation. Prior to the experiment, participants fasted for eight hours and underwent a preoperative evaluation. Propofol was administered through a target-controlled intravenous bolus and infusion, following the Marsh pharmacokinetic model, to achieve deep sedation. The effect-site concentration (ESC) was adjusted in 0.4 μg/mL increments until subjects exhibited no behavioral reactions, with the target concentration maintained for an average of 21.6 ± 10.2 min. Behavioral responsiveness was assessed using a rubber ball squeeze task, and vital signs were monitored regularly. MRI scans were conducted using a 3T Philips MRI scanner with a 32-channel transmit/receive head coil. Functional brain images were obtained using a gradient-echo EPI sequence with parameters: 28 slices, TR/TE = 800/25 ms (MB factor = 4), slice thickness = 4 mm, in-plane resolution = 3.4 × 3.4 mm, FOV = 220 mm, flip angle = 76°, and 64 × 64 image matrix. For six subjects scanned before hardware upgrades, the following parameters were used: 21 slices, TR/TE = 800/25 ms (MB factor = 3), slice thickness = 6 mm.

Dataset-2 included 30 healthy participants who also provided informed consent and were compensated. Participants were excluded if they had any contraindications for MRI, significant medical history, or other conditions that could affect the study. Propofol was manually titrated to achieve target concentrations for deep sedation, using a stepwise increase in ESC to identify the threshold for loss of responsiveness. The ESC was adjusted in increments (e.g., 1.5, 2.0, 2.5, and 3.0 μg/mL) with each target concentration maintained for 4 min. Once loss of responsiveness was achieved, the ESC was set one level higher than the threshold (e.g., if loss of responsiveness occurred at 2.0 μg/mL, ESC was maintained at 2.5 μg/mL) for approximately 32 min to minimize motion artifacts. In rare cases where participants remained responsive at 3.0 μg/mL, the ESC was increased to a maximum of 4.0 μg/mL. Behavioral responsiveness was assessed using a rubber ball squeeze task, and vital signs were monitored regularly. MRI scans were similarly conducted using a 3T Philips MRI scanner with a 32-channel transmit/receive head coil. Functional brain images were obtained using a gradient-echo EPI sequence with parameters: 40 slices, TR/TE = 1400/30 ms (MB factor = 4), slice thickness = 2.9 mm, in-plane resolution = 2.75 × 2.75 mm, FOV = 220 mm, flip angle = 76°, and 80 × 80 image matrix. The analysis excluded three participants due to excessive movement (frame-wise displacement > 0.8 mm in more than 25% of the data). Another participant was omitted due to the scanner’s technical issue.

#### 2.1.4. Propofol General Anesthesia

This dataset has been previously published in an investigation with distinct hypotheses and analyses [55]. The study received ethical clearance from the Institutional Review Board of Huashan Hospital, Fudan University. It enlisted 26 right-handed individuals (12 males and 14 females, aged between 27 and 64 years), all classified under the American Society of Anesthesiologists physical status I or II and scheduled for elective surgery to remove pituitary microadenomas. After excluding three individuals due to excessive movement, data from 23 participants were used for this study.

Participants abstained from solid foods for a minimum of eight hours and from liquids for two hours prior to the study. An intravenous catheter was inserted, and propofol was administered using a target-controlled infusion system, calibrated to maintain a steady effect-site concentration based on the Marsh model. In addition to general anesthesia, remifentanil at a dose of 1.0 μg/kg and succinylcholine at 1.5 mg/kg were given to facilitate endotracheal intubation. The propofol infusion started at 1.0 mg/mL with incremental adjustments of 0.1 mg/mL until reaching the target concentration of 4.0 μg/mL. The target concentration was maintained for a steady effect. Behavioral responsiveness was evaluated using the Ramsay scale. Participants were categorized as deeply sedated or anesthetized (Ramsay scores 5–6) when they did not respond to the verbal command “strongly squeeze my hand!” given twice during each evaluation phase. Under general anesthesia, ventilation was assisted using intermittent positive pressure, maintaining a tidal volume between 8–10 mL/kg and a respiratory rate of 10–12 breaths per minute. The entire procedure was supervised by two qualified anesthesiologists. Participants wore earplugs and headphones for auditory isolation and comfort during the fMRI scans.

Three 8 min fMRI scans were conducted during conscious baseline, light sedation, and general anesthesia. This study specifically utilized the fMRI data collected during the general anesthesia condition. Imaging was performed using a Siemens 3T MAGNETOM scanner with a standard 8-channel head coil, capturing whole-brain gradient-echo EPI images with the following settings: 33 slices, TR = 2000 ms, TE = 30 ms, slice thickness = 5 mm, FOV = 210 mm, flip angle = 90°, and a 64 × 64 image matrix. High-resolution anatomical images were also obtained.

#### 2.1.5. Lysergic Acid Diethylamide (LSD)

This dataset was sourced from an open-access database available on the OpenNEURO repository, involving 20 participants [56]. One participant was unable to complete the BOLD scans due to anxiety and the desire to exit the scanner, and four others were excluded from the group analyses due to excessive head movement. Consequently, 15 participants were included in the final analysis.

Volunteers participated in two sessions, receiving either a placebo or LSD in a counterbalanced order. A medical doctor inserted and secured a cannula in a vein in the antecubital fossa. Each participant received 75 μg of LSD intravenously via a 10 mL solution infused over two minutes, followed by a saline infusion. MRI scanning began approximately 70 mins after dosing, capturing changes associated with peak intensity between 60 and 90 mins post-administration. The two sessions for each participant were treated as individual data points.

Imaging was conducted on a 3T GE HDx system. Whole-brain functional images were acquired using a gradient-echo EPI pulse sequence with the following parameters: 35 slices, TR/TE = 2000/35 ms, slice thickness = 3.4 mm, field of view = 220 mm, image matrix = 64 × 64, flip angle = 90°, and a scan time of 7 min. High-resolution anatomical images were also obtained for co-registration with resting-state fMRI data.

#### 2.1.6. Ketamine

This dataset was previously published with different analyses [15,57]. The study received approval from the Institutional Review Board of Huashan Hospital, Fudan University, and informed consent was obtained from all participants. Twelve right-handed volunteers were recruited, all classified as American Society of Anesthesiologists physical status I or II, and with no history of brain or major organ dysfunction, or use of neuropsychiatric drugs.

Ketamine was administered through an intravenous catheter in the left forearm. fMRI scanning spanned the entire experiment, lasting between 44 and 62 min (mean 54.6 ± 5.9 min). Initially, a 10- min baseline conscious condition was recorded (except for two participants with baseline conditions of 6 and 11 min). Ketamine was infused at 0.05 mg/kg per minute for 10 min (0.5 mg/kg total), followed by 0.1 mg/kg per minute for another 10 min (1.0 mg/kg total), with two participants receiving only the latter infusion. The infusion was then discontinued, and participants spontaneously regained responsiveness. Two certified anesthesiologists were present throughout, with resuscitation equipment available. Participants wore earplugs and headphones during fMRI scanning. The protocol included the period before and during ketamine-induced loss of responsiveness, but only the period before loss of responsiveness was analyzed for this study, focusing on subanesthetic psychedelic experiences.

A Siemens 3T MAGNETOM scanner with a standard eight-channel head coil was used. Whole-brain functional images were acquired using a gradient-echo EPI pulse sequence with the following parameters: 33 slices, TR/TE = 2000/30 ms, slice thickness = 5 mm, field of view = 210 mm, image matrix = 64 × 64, flip angle = 90°, and a scan time of 12 min. High-resolution anatomical images were also obtained for co-registration with resting-state fMRI data.

#### 2.1.7. Nitrous Oxide (N_2_O)

This dataset has been previously published using different analyses [58]. There were 17 healthy participants who underwent two resting-state fMRI scans before and during exposure to subanesthetic levels of nitrous oxide (35% concentration). The study was conducted at the University of Michigan Medical School and approved by the Institutional Review Board (HUM00096321). Participants provided written informed consent and were thoroughly informed of the study’s potential risks and benefits. One subject was excluded due to excessive head movement.

The study involved a pre-scan visit to brief participants on the protocol and a scanning visit within three days. During the scanning visit, fMRI data were collected during both placebo and subanesthetic nitrous oxide inhalation, with nitrous oxide administered to achieve at least 5 min of equilibrium. Ondansetron (4–8 mg IV) was provided to mitigate the common side effect of nausea, with additional medications available if needed. Standard intraoperative monitoring devices, including electrocardiogram, blood pressure, pulse oximetry, and capnography, were used throughout the experiment. Participants wore earplugs and headphones during the fMRI scans to reduce interference from external stimuli.

Imaging data were acquired using a 3T Philips Achieva MRI scanner at Michigan Medicine, University of Michigan. Functional whole-brain images were obtained using a T2*-weighted echo-planar sequence with the following parameters: 48 slices, TR/TE = 2000/30 ms, slice thickness = 3 mm, field of view = 200 × 200 mm, flip angle = 90°, and a scan time of 6 min. High-resolution anatomical images were also acquired for co-registration with the resting-state fMRI data.

#### 2.1.8. Neuropsychiatric Disorders

The dataset was sourced from the OpenNEURO database, provided by the UCLA Consortium for Neuropsychiatric Phenomics [59]. The study received approval from the Institutional Review Boards at UCLA and the Los Angeles County Department of Mental Health. Informed consent was obtained from all participants, who were compensated post-experiment. The original dataset included 272 participants, comprising healthy individuals (*n* = 122) and individuals with psychiatric disorders including ADHD (*n* = 40), bipolar disorder (*n* = 49), and schizophrenia (*n* = 50). Healthy groups were randomly divided into three subgroups and used as control groups (*n* = 41 for ADHD and bipolar disorder; *n* = 40 for schizophrenia). Detailed population characteristics are available in reference.

Neuroimaging was performed using a 3T Siemens Trio scanner. Functional MRI data were obtained with a T2*-weighted EPI sequence (slice thickness: 4 mm, 34 slices, TR: 2 s, TE: 30 ms, flip angle: 90°, matrix: 64 × 64, FOV: 192 mm). Data were excluded if T-1 images or resting-state data were missing, if head motion exceeded 3 mm, or if the degrees of freedom were insufficient after motion scrubbing and band-pass filtering.

### 2.2. Data Preprocessing

Preprocessing steps were conducted using AFNI (linux_ubuntu_16_64; http://afni.nimh.nih.gov/ (accessed on 27 August 2024)), except for the LSD dataset. The steps included: (1) discarding the first two fMRI frames of each scan; (2) performing slice timing correction; (3) correcting for rigid head motion/realignment, with frame-wise displacement (FD) defined as the Euclidean norm of the six-dimensional motion derivatives. Frames where the derivative value exceeded an FD of 0.4 mm, along with the preceding frame, were excluded; (4) co-registering with T1 anatomical images; (5) spatially normalizing into Talairach stereotactic space; (6) using AFNI’s function 3dTproject to band-pass filter the time-censored data to 0.01–0.1 Hz, removing undesired components such as linear and nonlinear drift, head motion time series and their derivatives, and mean time series from the white matter and cerebrospinal fluid via linear regression; (7) applying spatial smoothing with a 6 mm full-width at half-maximum isotropic Gaussian kernel; (8) normalizing each voxel’s time-course to zero mean and unit variance.

For LSD dataset, preprocessing steps included: (1) removing the first three volumes; (2) de-spiking; (3) slice time correction; (4) motion correction; (5) brain extraction; (6) rigid body registration to anatomical scans; (7) non-linear registration to a 2 mm MNI brain; (8) scrubbing using an FD threshold of 0.4; (9) spatial smoothing with a 6 mm kernel; (10) band-pass filtering between 0.01 and 0.08 Hz; (11) linear and quadratic de-trending; and (12) regressing out undesired components such as motion-related parameters.

GSR can minimize unwanted global confounds such as low-frequency respiratory volume and cardiac rate. While the GSR procedure is known to have minimal effects on the results of cortical gradient analysis [46], it significantly influences functional connectivity profile and alters graph measures. Therefore, for connectivity and graph measures, we performed calculations using both GSR and noGSR signals to ensure robustness.

After preprocessing, fMRI time courses were extracted from 400 predefined cortical areas (each assigned to one of seven predefined subnetworks) using a well-established brain parcellation scheme [60,61]. Seven predefined networks include visual (VIS), somatomotor (SMN), dorsal attention (DAN), ventral attention (VAN), limbic (LIM), frontoparietal (FPN), and default-mode (DMN) networks [60].

### 2.3. Feature Extraction

#### 2.3.1. Connectivity Measures

Functional connectivity matrix (400 × 400) was calculated by Pearson correlation of the time courses. We then calculated the mean and standard deviation of functional connectivity values. Mean and standard deviation calculations were performed across the entire matrix, within 7 predefined subnetworks, and between 21 pairs of subnetworks (i.e., the entries corresponding to connections between network A and network B, A ≠ B). This approach resulted in a total of 58 features (29 (1 whole brain + 7 subnetworks + 21 between-network pairs) × 2 (mean and standard deviation)). This procedure was independently applied to noGSR and GSR data, resulting in two separate sets of features.

#### 2.3.2. Graph Measures

Eight unweighted brain graphs (whole brain and seven subnetworks) were constructed after binarizing the functional connectivity matrices at 90% sparsity (10% edge density), preserving the top 10% of edges, which is consistent with the common practice in the literature [62,63]. Four graph-theoretic measures were calculated: efficiency (mean of inverse distance), clustering coefficient, modularity, and mean participation coefficient. The calculations are as follows: (1) Efficiency was calculated on two levels. First, a 400 × 400 inverse distance matrix was created from the whole-brain binary connectivity matrix using the ‘distance_bin’ function in the Brain Connectivity Toolbox, yielding 29 features (1 whole-brain average, 7 intra-network, and 21 inter-network pairs). Second, efficiency for the seven subnetworks was calculated from smaller network-level connectivity matrices (denoted ‘Intra-efficiency’). (2) Clustering coefficient was also calculated on two levels. First, local clustering coefficients were derived from the whole-brain functional connectivity matrix using the ‘clustering_coef_wu’ function in BCT, extracting eight features (one whole-brain average and seven intra-network averages). Second, global clustering coefficients for the network-level matrices of the seven networks were calculated (denoted ‘Intra-clustering’), adding seven more features (total fifteen features). (3) Modularity was calculated using the ‘modularity_und’ function, resulting in eight values, one for each brain graph. (4) Participation coefficient was also calculated on two levels. From the whole-brain graph, node-wise participation coefficient was calculated using the ‘participation_coef’ function, extracting eight features (one whole-brain average and seven intra-network averages). Additionally, mean participation coefficients for the network-level matrices of the seven networks were calculated (denoted ‘Intra-participation’), adding seven more features (total fifteen features). Newman’s algorithm, with gamma set to 1, was used for community detection needed for modularity and participation coefficient calculations. This procedure resulted in a total of 74 features. It was independently applied to both noGSR and GSR data, resulting in two separate sets of features. All calculations were conducted using MATLAB functions in the Brain Connectivity Toolbox [44].

#### 2.3.3. Cortical Gradient Analysis

A 400 × 400 functional connectivity matrix generated from GSR signal was used for cortical gradient calculation [46], computed using the BrainSpace toolbox v0.1.10 (https://brainspace.readthedocs.io/en/latest/ (accessed on 27 August 2024)) [64]. Following prior studies, we *z*-transformed and binarized the connectivity matrix at 90% sparsity, retaining the top 10% of weighted connections per row [45,46,65,66]. A normalized cosine angle affinity matrix was then calculated to capture the similarity of connectivity profiles between cortical areas. Gradient components were identified using a diffusion map embedding algorithm, with parameters α set at 0.5 to control the influence of the density of sampling points and *t* set to 0 for automated diffusion time estimation [45,46,65,66]. These settings preserved global relations between data points. In result, cortical areas with similar functional connectivity profiles are positioned near each other on the gradient map, while areas with more dissimilar connectivity profiles are located farther apart.

Using Procrustes rotation, group-level gradient solutions were aligned to a subsample of the Human Connectome Project dataset (*n* = 217) from the BrainSpace toolbox. This alignment ensured stability and comparability of gradients across individuals, addressing variations in eigenvector orderings and sign ambiguities. We retained the first three cortical gradients (400 × 3 values) to form a virtual 3D gradient space.

Aligned with previous studies [46,67], we computed four types of features to quantify individual cortical gradient profiles: range, dispersion, network eccentricity, and between-network distance. The calculations were as follows: (1) Range: For both the whole brain and seven predefined subnetworks, we determined the numerical range for each gradient, defined as the distance from the minimum to maximum gradient eigenvector values ([1 whole brain + 7 subnetworks] × 3 dimensions = 24 features). (2) Dispersion: Network dispersion for the eight networks (one whole brain + seven subnetworks) was quantified as the sum squared Euclidean distance of all regions assigned to a given network to the network centroid within the 3D gradient space (eight features). (3) Network eccentricity: Calculated as the squared Euclidean distance between the centroids of a given network and the whole-brain network (seven features). (4) Between-network distance: Calculated for every pair of subnetworks as the squared Euclidean distance between the centroids of the two networks (21 pairs among 7 networks = 21 features). In total, 60 features were extracted from the cortical gradient analysis.

### 2.4. Classification

#### 2.4.1. Base Models

The feature extraction method produced five sets of features, each containing 58–74 features (Figure 1B). An SVM with a linear kernel was employed for training on each feature set using ‘fitcsvm’ function of MATLAB. To ensure rigorous model evaluation, a nested cross-validation approach was used, comprising outer 10-fold and inner 3-fold cross-validation loops (Figure 1C).

In the outer loop, the dataset was divided into 10 folds. For each iteration, one fold served as the test set, while the remaining nine folds formed the outer training set. Each feature set underwent standardization within this outer training set. The standardized data were then subjected to the inner 3-fold cross-validation loop for hyperparameter tuning. The SVM model was trained on the inner training data and validated on the inner validation data to identify the optimal hyperparameters and feature weights. Hyperparameters were tuned within the ranges: box constraint (10^−6^ to 10^3^) and kernel scale (10^−4^ to 10^2^). The set of hyperparameters that minimized the average loss across the three inner folds was selected.

After determining the optimal hyperparameters, five base SVM models were trained on the outer training set, which included both the inner training and validation sets. These optimized models were then evaluated on the outer test set. The entire procedure was repeated 50 times to account for the randomness in cross-validation partitioning.

#### 2.4.2. Ensemble Models

We explored three different methods for constructing ensemble models to improve classification performance and robustness. The methods are as follows: (1) Feature integration: We concatenated all features from the five individual feature sets into a single comprehensive feature set, resulting in a total of 324 features. A single SVM model was then trained on this combined feature set. This method leverages the full breadth of information available across all feature sets. (2) Hard voting: During prediction, each base model provided a class prediction, and the final class label was determined by majority vote. (3) Soft voting: The prediction scores from each base model were summed to produce a final score for each class. By combining different feature sets and prediction strategies, we aimed to create a more robust classification framework that could better generalize for unseen data.

#### 2.4.3. Model Evaluation

During hyperparameter tuning, the hinge loss function was employed to optimize the SVM models within the inner cross-validation loops. For the final evaluation on the test set, we used two performance metrics: balanced accuracy and the area under the receiver operating characteristic curve (AUC). Balanced accuracy (average of sensitivity and specificity) accounts for the class imbalance, providing a more equitable measure of model performance across different classes.
(1)$$\mathrm{Balanced}\;\mathrm{accuracy}=\frac{1}{2}\left(\mathrm{Sensitivity}+\mathrm{Specificity}\right)=\frac{1}{2}\left(\frac{\mathrm{TP}}{\mathrm{TP}+\mathrm{FN}}+\frac{\mathrm{TN}}{\mathrm{TN}+\mathrm{FP}}\right)$$
Optimal score threshold that maximizes the balanced accuracy was searched within the range of −1 and +1.

AUC, on the other hand, evaluates the model’s ability to discriminate between classes by summarizing the trade-off between the true positive rate and false positive rate across different threshold settings.

#### 2.4.4. Feature Importance Evaluation

To determine feature importance, we utilized the linear kernel function for our SVM models, which allows for straightforward interpretation of feature weights. The importance of each feature was quantified using the absolute value of its corresponding beta coefficient from the SVM models trained with all 324 features. This approach leverages the linear relationship between features and the decision boundary in the SVM, with higher magnitude of beta coefficients indicating greater importance. To obtain a robust estimate of feature importance, we averaged the absolute coefficients across the entire 1500 models (3 conditions per meta-condition × t10 outer folds × 50 repetitions) within each meta-condition.

## 3. Results

### 3.1. General Model Performance

Figure 2 presents the balanced accuracy of both base models and various ensemble models aggregated across nine conditions. All models achieved a balanced accuracy exceeding 70%. There was a significant difference in balanced accuracy among eight models (two-way ANOVA, *p* < 0.0001). Among the five base models, the model utilizing connectivity features from GSR matrix demonstrated the highest performance, with a balanced accuracy of 76.4%. This was followed by the models using noGSR connectivity (74.2%), GSR graph (73.3%), noGSR graph features (70.5%), and cortical gradient (70.1%). In the evaluation of ensemble models, the soft voting method yielded the highest balanced accuracy at 78.7%, significantly outperforming other ensemble models and all base models (*p* < 0.0001, Bonferroni correction). The feature integration model achieved a balanced accuracy (76.7%) slightly below that of soft voting, followed by the hard voting model (74.4%). The results of hyperparameter tuning are provided in Table 2.

### 3.2. Condition-Specific Model Performance

Figure 3 illustrates the condition-specific performances of both base and ensemble models across three meta-conditions: unconscious, psychedelic, and neuropsychiatric. Following the findings of Figure 2, we selected soft voting as the representative ensemble model here.

Each base model demonstrated unique performance patterns across different conditions. The GSR connectivity model excelled in the highest number of conditions (four out of nine), showing top performance for N2 sleep, LSD, bipolar disorder, and schizophrenia. The noGSR connectivity model performed best in three conditions: deep sedation, general anesthesia, and ADHD. The gradient model and GSR graph model each showed highest performance in one condition (ketamine and N_2_O, respectively). The noGSR graph model did not outperform other models in any condition. This indicates that different types of features contribute uniquely to the classification performance.

Among the three meta-conditions, the models demonstrated the best performance at unconscious condition, with accuracy of soft voting models exceeding 75% (N2 sleep: 76.8%; deep sedation: 97.9%; general anesthesia: 97.2%) and AUC values exceeding 0.8 (N2 sleep: 0.82; deep sedation: 1.00; general anesthesia: 0.99) across all three sub-conditions (Figure 3A,B). The models achieved near-perfect performance for the deep sedation and general anesthesia conditions, both with AUC values approximately equal to 1. All five base models showed comparable performance levels, with the ensemble model demonstrating a marginally higher performance for deep sedation and general anesthesia. Confusion matrices for these conditions (Figure 3G–I) show the robustness of the models.

In the psychedelic meta-condition, there was a notable variation in model performance across the different conditions (Figure 3C,D). LSD condition yielded the highest accuracy of 91.1% and AUC of 0.94. Ketamine (accuracy of 72.0% and AUC of 0.65) and N_2_O (accuracy of 69.3% and AUC of 0.66) models showed similar performance. The confusion matrices for these conditions (Figure 3J–L) highlight the variation in model performance.

The neuropsychiatric meta-condition showed relatively lower accuracy compared to the other two meta-conditions (Figure 3E,F). Among the three sub-conditions, schizophrenia achieved the highest accuracy of 71.9% and AUC of 0.76, followed by bipolar disorder (accuracy of 69.1% and AUC of 0.71) and ADHD (accuracy of 62.7% and AUC of 0.60). The confusion matrices (Figure 3M–O) reveal moderate true positive rates and higher false positive rates compared to the unconscious and psychedelic conditions.

### 3.3. Cross-Dataset Performance

Figure 4 illustrates the classification performance across various conditions. Each cell in the heatmaps represents the balanced accuracy or AUC score for a given pair of train-test conditions. Importantly, test datasets had their own baselines, allowing us to assess the model performance in distinguishing between baseline and atypical brain states within a new population.

It is important to emphasize that these cross-dataset evaluations do not measure the model’s ability to differentiate between the two atypical brain states in the train and test datasets. Instead, they assess the model’s capacity to classify the test dataset into two groups based on the features it learned during training. For example, if a model trained on N2 sleep data shows high accuracy when tested on general anesthesia data, it does not mean it is directly differentiating between sleep and anesthesia. Rather, it suggests that the features the model learned to distinguish normal wakefulness from sleep are also somewhat effective in distinguishing normal wakefulness from general anesthesia.

Models trained on unconscious meta-conditions showed high transferability within its sub-conditions (balanced accuracy: 66.8 ± 7.2%; AUC: 0.80 ± 0.05; mean ± SEM). Models trained on ketamine and LSD, as well as those trained on bipolar disorder and schizophrenia, also demonstrated relatively high transferability (accuracy > 0.63 and AUC > 0.70) to each other. High performance was observed also across meta-conditions. In particular, neuropsychiatric models showed high performance on unconscious test sets (accuracy: 63.0 ± 7.6%; AUC: 0.74 ± 0.11), although the reverse was not true (accuracy: 54.5 ± 6.0%; AUC: 0.57 ± 0.08). This was also the case for the deep sedation model on the N_2_O dataset (accuracy: 67%; AUC: 0.84) and the bipolar disorder model on the LSD dataset (accuracy: 63%; AUC: 0.88).

Several train–test pairs exhibited balanced accuracy or AUC scores below 0.5, indicating possible opposite characteristics between these conditions. For instance, models trained on ketamine and LSD showed performances lower than 0.5 when tested on N2 sleep and deep sedation datasets. Similarly, models trained on ADHD and tested on bipolar disorder and schizophrenia, and vice versa, displayed performance metrics lower than 0.5, suggesting distinct differences between these conditions.

### 3.4. Feature Importance

We explored the relative importance among the 324 features used for our classification models. Since we used linear kernel function, we could quantify feature importance from the weight of the vector orthogonal to the hyperplane. Therefore, the normalized absolute value of beta coefficients from the feature-integrated ensemble models was used as a proxy for feature importance (Figure 5). Recognizing that feature importance could vary across meta-conditions, we analyzed it separately for each meta-condition. As shown in Figure 5A–C, both noGSR and GSR-derived features were present in the top 10 list.

In the case of the unconscious meta-condition (Figure 5A), the most important feature was the mean of visual-somatomotor connectivity (both in GSR and noGSR), followed by the dispersion of the somatomotor network in cortical gradient and the mean participation coefficient of the visual network. Notably, eight out of ten features were related to unimodal networks (visual and somatomotor networks). Additionally, features derived from gradient, connectivity, and graph measures appeared without a specific preference.

Figure 5B presents the feature importance for the psychedelic meta-condition. The highest importance was assigned to whole-brain dispersion, which greatly surpassed the other features. This was followed by the mean participation coefficient and the intra-clustering of the ventral attention network (see Methods for the definition of intra-clustering). Unlike the unconscious condition, the top 10 list consisted of attention, limbic, and transmodal networks. Visual network did not appear on the list.

The feature importance for the neuropsychiatric meta-condition is shown in Figure 5C. The leading features were the mean participation coefficient and efficiency of the limbic network, followed by the intra-clustering of the ventral attention network. The majority (seven out of ten) of the important features were graph-theoretical metrics. Most of these features were associated with unimodal, attention, and limbic networks.

## 4. Discussion

In this study, we demonstrated an approach for classifying baseline and atypical brain states that can be generalized to various conditions, including unconscious, psychedelic, and neuropsychiatric conditions. Our method integrates a wide range of features, including connectivity measures, graph-theoretic metrics, and cortical gradients. The soft voting ensemble model performed best, highlighting the advantage of combining multiple prediction strategies. Our models showed varying degrees of transferability across different datasets, with performance dependent on the specific brain states and feature sets used. These results suggest that our approach has potential for developing biomarkers for specific brain states, particularly in distinguishing unconscious and certain psychedelic states from normal wakefulness.

Incorporating multiple sets of features is a promising strategy for developing broadly applicable brain classification models. We employed three feature types aiming to provide unique insights into the brain’s functional configurations. Functional connectivity features show interactions between regions, graph metrics reveal complex network properties, and gradient features provide overall brain organization. Notably, the inclusion of cortical gradients as machine learning features is a novel approach, barely explored in previous research. By combining these diverse features, our models become more robust and adaptable across different conditions and populations. This approach also demonstrated high transferability across varied test conditions. Using features from both GSR-applied and GSR-unapplied data ensures the model is resilient to different preprocessing methods, making it more versatile for clinical and research applications.

Our analysis shows that models based on different feature sets perform differently across conditions (e.g., Figure 2 and Figure 3). This varied performance across models and conditions underscores the importance of using multiple feature types and preprocessing methods to capture the full spectrum of brain states. Models using GSR consistently outperformed noGSR models, suggesting GSR effectively reduces noise while preserving state-specific information [41]. Among the three feature types, connectivity features were most accurate, suggesting that interregional connectivity profiles are more informative in distinguishing brain states. This finding is encouraging for practical approaches to brain state classification, as connectivity measures are computationally more straightforward compared to gradient and graph-theoretical values. Nevertheless, the other two feature types also show superior performance under specific conditions, highlighting their unique strengths. For example, cortical gradient features were most informative in classifying the subanesthetic ketamine dataset. This suggests that gradient features might be particularly adept at classifying ketamine-induced psychedelic states, potentially due to their ability to capture the drug’s widespread effects on global brain organization and hierarchical processing, which might be less discernible using more localized connectivity or graph measures. The soft voting ensemble model performed best overall, as it combines predictions from multiple models, leveraging their collective strengths and mitigating individual weaknesses.

Among the three meta-conditions, model performance was superior in classifying unconscious states (e.g., Figure 3A,B). This is likely attributed to the substantial functional reconfigurations of the brain involved in these states (e.g., deep sedation and general anesthesia) compared to conscious wakefulness. This clear difference led to near-perfect classification for deep sedation and general anesthesia. N2 sleep, however, showed a relatively lower accuracy. N2 sleep represents a natural, physiological state of altered consciousness, unlike pharmacologically induced states like deep sedation and general anesthesia. Also, N2 is an intermediate stage of sleep, potentially retaining more similarities to wakefulness than deeper sleep stages or anesthesia [68]. Moreover, N2 sleep involves complex and dynamic patterns of brain activity that may vary more between individuals than drug-induced states [68,69]. There is also a possibility of preserved internal awareness during N2 sleep, albeit different from wakefulness or anesthesia [70]. In psychedelic conditions, LSD was most accurately classified, outperforming ketamine and N_2_O (e.g., Figure 3C,D). This may be because LSD has unique, strong effects on brain connectivity and network dynamics, creating more distinct patterns [71]. Collectively, the data suggest that the models functioned best for detecting canonical states within a meta-condition, with anesthesia inducing the canonical unconscious state and LSD inducing the canonical psychedelic state.

The neuropsychiatric meta-condition showed the lowest performance (Figure 3E,F), possibly due to the high variability and complexity of these disorders, making them harder to classify accurately. Among neuropsychiatric conditions, our models classified bipolar disorder and schizophrenia more accurately than ADHD (e.g., Figure 3N,O vs. Figure 3M). This difference may be because bipolar disorder and schizophrenia have more distinct and consistent neurobiological markers. These disorders often involve significant changes in brain connectivity and structure, which our models can more easily detect. In contrast, ADHD symptoms and associated brain patterns may fluctuate more over time, which likely contributes to the lower classification accuracy for this condition [72,73].

The performance of our models aligns with previous research findings. For instance, earlier studies on sleep stage detection using functional connectivity data reported approximately 80% accuracy, which is comparable to our results [30]. In the context of anesthesia, previous machine learning analyses using functional connectivity achieved an AUC of 0.67 [22], whereas our analysis significantly outperformed this, approaching nearly 100% AUC. For ADHD and bipolar disorder, typical studies report accuracy ranges from 60% to 80%, and our model performed within this expected range [74,75]. Regarding schizophrenia, a functional connectivity-based study using the same dataset as ours reported 70% accuracy [76], while a machine learning approach using graph theoretical data with the same dataset achieved 79% accuracy [77]. Our performance, with a balanced accuracy of 72%, is consistent with these findings. Moreover, it is important to highlight that our methodology has a distinct advantage in terms of generalizability, making it applicable across various brain states.

We investigated the performance of models trained on specific conditions and tested across various other conditions. For 57% of the total train–test pairs, the balanced accuracy exceeded 0.6, indicating some shared characteristics among the conditions. Notably, models trained on neuropsychiatric conditions demonstrated unexpectedly high accuracy when tested on unconscious conditions. This intriguing finding warrants further investigation, as neuropsychiatric disorders and unconscious states have distinct neurobiological bases. The observed cross-condition performance might reflect a sensitivity to deviations from normal brain function, rather than specific shared features. It is plausible that features learned to identify neuropsychiatric conditions also effectively detect broad changes in unconscious states, despite differing underlying neural mechanisms.

It is important to note that the varying classification accuracies observed across brain states may reflect more than just model performance; they could be indicative of a continuum of conscious states [50,78,79,80]. For instance, the high accuracy in classifying general anesthesia compared to the lower accuracy for N2 sleep might not signify a model limitation, but rather the nuanced nature of consciousness. General anesthesia represents a more profound loss of consciousness, while N2 sleep may retain some elements of conscious experience [70]. Similarly, the differential performance in classifying various psychedelic states could reflect their varying positions along a gradient of altered consciousness, rather than discrete categories [57,81]. This interpretation aligns with emerging views of consciousness as a multidimensional continuum rather than binary states [80], with variations along spectrums like arousal, connectedness, and awareness [79]. Future research should explore how these classification accuracies might map onto proposed multidimensional models of consciousness.

Intriguingly, we observed contrasting relationships between certain brain states. Notably, we observed an inverse relationship between LSD and both N2 sleep and deep sedation. This aligns with the existing literature suggesting LSD induces hyperintegrated brain patterns, while N2 sleep and sedation exhibit hypersegregated connectivity [82,83,84]. This indicates that the models perceive sedation and sleep as the opposite or “mirror image” of a psychedelic state.

Similarly, ADHD showed an opposite relationship with bipolar disorder and schizophrenia. This may be due to differences in affected brain networks. ADHD mainly impacts attention and executive function areas, often showing hypoconnectivity, especially in prefrontal regions [85]. In contrast, bipolar disorder and schizophrenia involve widespread changes across multiple networks, including the default mode, salience, and cognitive control systems. They often display complex patterns of brain connectivity. These fundamental differences in neural patterns likely explain why models trained on one condition perform poorly when classifying the others, indicating distinct underlying neural mechanisms.

We explored the relative importance of various features used in our classification models. For the unconscious meta-condition (e.g., Figure 5A), the most crucial features were linked to unimodal networks, highlighting the likely shift to unimodal-dominant networks in unconscious states. This likely reflects the significant changes in sensory processing and motor function that occur during diminished consciousness, such as in deep sedation and general anesthesia.

In the psychedelic meta-condition (e.g., Figure 5B), whole-brain dispersion emerged as the most important feature, significantly outweighing all others. This suggests that psychedelic states are best characterized by global changes in gradient space, reflecting the widespread effects of psychedelic substances on brain network dynamics. The absence of unimodal networks among the top features in the psychedelic meta-condition is intriguing and potentially significant. This suggests psychedelic substances primarily affect higher-order cognitive processes and integrative brain functions, rather than basic sensory and motor processes. Psychedelics are known to alter perception, cognition, and self-awareness, which are primarily associated with transmodal and associative networks [86].

For the neuropsychiatric meta-condition (e.g., Figure 5C), graph-theoretical metrics accounted for 7 out of the 10 most important features. This suggests that complex network properties are key in identifying neuropsychiatric disorders, which aligns with our understanding that these conditions often involve widespread disruptions in brain network organization. The importance of unimodal (sensorimotor) and limbic networks highlights their crucial role in sensory processing, emotional regulation, and cognitive function.

Overall, these differences in feature importance across meta-conditions suggest that the underlying neural mechanisms vary significantly, requiring tailored approaches for accurate classification. This underscores the value of our feature-integrated ensemble models, which leverage the strengths of multiple feature types to achieve robust and generalizable performance across different brain states.

Our study has limitations. First, the sample size for each condition is relatively small (*n* ≤ 50). Larger datasets would be necessary to further verify the robustness and reliability of the results. The exact feature importance order displayed in this study may vary due to the high ratio of features to the sample size, which is not ideal and could lead to overfitting or unstable feature rankings. Second, our study did not investigate node-level features, focusing instead on aggregated network-level metrics. Node-level analysis, in combination with appropriate feature selection techniques, could provide more granular insights into specific brain regions contributing to the classification. Third, our study did not account for potential scanner effects. The data were collected from multiple MRI scanners, which could introduce variability. Addressing scanner-specific artifacts and harmonizing data across different scanners might be important steps to enhance the robustness of the models [87]. Lastly, the reliability and generalizability of our method varied across conditions, indicating a need for further validation for clinical applications.

## 5. Conclusions

Our study introduces a novel ensemble modeling approach integrating diverse neuroimaging features—cortical gradients, connectivity measures, and graph-theoretic metrics—to classify brain states, achieving an average balanced accuracy of 79% (ranging from 62% to 98%), significantly outperforming individual base models (70–76%). This approach offers valuable insights into neural signatures, paving the way for developing biomarkers and advancing our understanding of brain states across various conditions.

## Figures and Tables

**Figure 1 brainsci-14-00880-f001:**
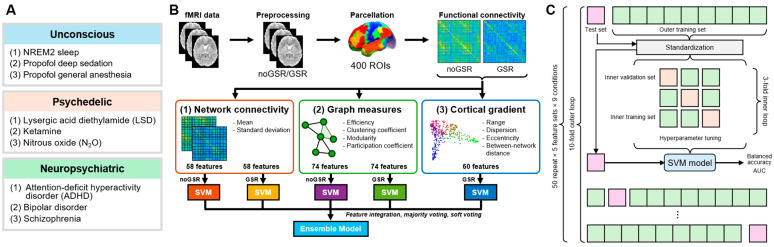
Analysis procedure of brain state classification based on resting-state fMRI. (**A**) Three meta-conditions and their sub-conditions. (**B**) Schematics of model constructure procedure. Resting-state fMRI data were preprocessed and parcellated to form two 400 × 400 functional connectivity matrices (noGSR and GSR). Three types of features were extracted from network connectivity, graph measures, and cortical gradient. Five base models trained with each set of features were stacked as an ensemble model. (**C**) Repeated nested cross-validation procedure. For five feature sets, data are split into 10 and 3 folds for outer and inner cross-validation loops, respectively. The resulting SVM model was tested on the 10% test sets, with balanced accuracy (average of specificity and sensitivity) and AUC as evaluation metrics. This whole process is repeated 50 times for each training condition. GSR: global signal regression; ROI: region of interest; SVM: support vector machine.

**Figure 2 brainsci-14-00880-f002:**
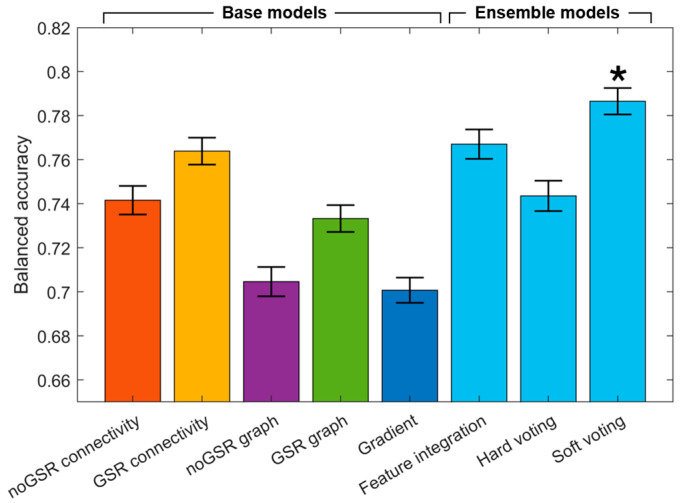
Average accuracy of various models. Balanced accuracy was calculated from five base models and three ensemble models generated from various stacking methods. Asterisks indicate significant differences compared to the other seven models. Error bars denote mean ± SEM.

**Figure 3 brainsci-14-00880-f003:**
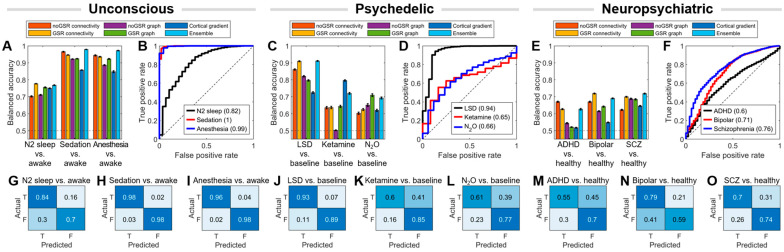
Balanced accuracy, area under the receiver operating characteristic curve (AUC), and confusion matrix across all conditions. (**A**,**C**,**E**) Balanced accuracy of five base models and the soft-voting ensemble model for (**A**) unconscious vs. awake, (**C**) psychedelic vs. baseline, and (**E**) neuropsychiatric vs. healthy conditions. Error bars denote mean ± SEM. (**B**,**D**,**F**) AUC of ensemble models for (**B**) unconscious, (**D**) psychedelic, and (**F**) neuropsychiatric meta-conditions. Numbers in the parentheses are AUC values. (**G**–**O**) Normalized confusion matrix for all sub-conditions. T and F denote atypical and baseline (healthy awake) conditions, respectively. LSD: lysergic acid diethylamide; ADHD: attention-deficit hyperactivity disorder; SCZ: schizophrenia.

**Figure 4 brainsci-14-00880-f004:**
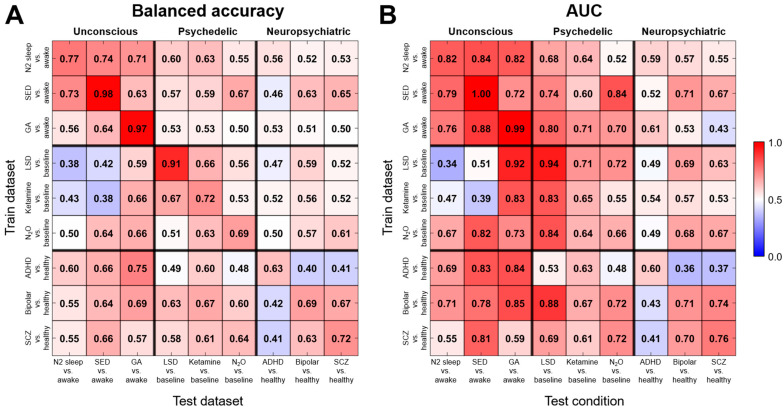
Cross-dataset classification performance. (**A**) Balanced accuracy and (**B**) AUC of models trained on a specific condition (rows) tested with various test conditions (columns). Each cell represents the balanced accuracy or AUC score for a given pair of train–test conditions. Diagonal entries are 10-fold cross-validation performance. SED: propofol deep sedation; GA: general anesthesia; LSD: lysergic acid diethylamide; ADHD: attention-deficit hyperactivity disorder; SCZ: schizophrenia; AUC: area under the receiver operating characteristic curve.

**Figure 5 brainsci-14-00880-f005:**
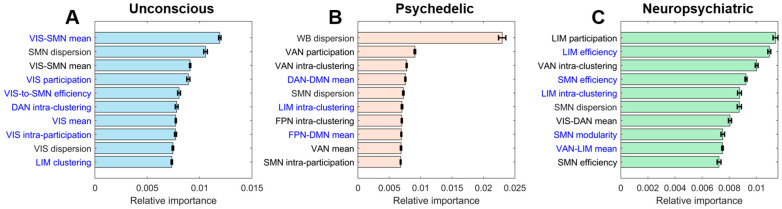
Top 10 average feature importance of the feature-integrated ensemble models trained for (**A**) unconscious, (**B**) psychedelic, and (**C**) neuropsychiatric meta-conditions. Feature importance was quantified by the normalized absolute beta coefficients from the SVM models. Features labeled with blue are derived from the noGSR connectivity matrix. Error bars denote mean ± SEM. WB: whole-brain; VIS: visual network; SMN: somatomotor network; DAN: dorsal attention network; VAN: ventral attention network; LIM: limbic network; FPN: frontoparietal network; DMN: default-mode network.

**Table 1 brainsci-14-00880-t001:** A summary of the datasets used in this study.

Dataset	Subjects (Male/Female)	Age (Year)	Scanner, TR (s)	Time Points
NREM2 sleep	29 (17/16)	22.1 ± 3.2	3 T, 2.1	2971 ± 833
Propofol deep sedation-1	26 (13/13)	25.0 ± 4.1	3 T, 0.8	8370
Propofol deep sedation-2	26 (9/17)	24.3 ± 5.2	3 T, 1.4	5576
Propofol general anesthesia	26 (12/14)	47.0 ± 11.4	3 T, 2.0	480
LSD	15 (10/5)	38.4 ± 8.6	3 T, 2.0	868
Ketamine	12 (7/5)	46.8 ± 13.4	3 T, 2.0	1628 ± 189
N_2_O	16 (8/8)	24.6 ± 3.7	3 T, 2.0	360
Neuropsychiatric disorders	272 (155/117) (healthy: 130; ADHD: 43; bipolar: 49; schizophrenia: 50)	33.0 ± 9.2	3 T, 2.0	150

**Table 2 brainsci-14-00880-t002:** Hyperparameter tuning results. The most frequently chosen values are provided.

**Condition**	**Best Hyperparameter (Box Constraints/Kernel Scale)**					
	**noGSR Connectivity**	**GSR Connectivity**	**noGSR Graph**	**GSR Graph**	**Gradient**	**Feature Integration**
NREM2 sleep	10^−5^/10^−3^	10^−5^/10^−3^	10^−5^/10^−3^	10^−6^/10^−3^	10^−5^/10^−3^	10^−6^/10^−3^
Propofol deep sedation	10^−5^/10^−3^	10^3^/10^−3^	10^−6^/10^−4^	10^−5^/10^−3^	10^−5^/10^−3^	10^3^/10^−3^
Propofol general anesthesia	10^2^/10^−3^	10^3^/10^−3^	10^3^/10^−4^	10^−5^/10^−1^	10^−5^/10^−3^	10^3^/10^−3^
LSD	10^−2^/10^−1^	10^−6^/10^−4^	10^3^/10^−2^	10^−6^/10^2^	10^0^/10^0^	10^−6^/10^−3^
Ketamine	10^−6^/10^−3^	10^3^/10^−4^	10^−6^/10^−4^	10^3^/10^0^	10^3^/10^−4^	10^3^/10^−3^
N_2_O	10^−6^/10^−3^	10^−6^/10^−4^	10^−6^/10^−4^	10^−6^/10^−4^	10^3^/10^−4^	10^−6^/10^−3^
ADHD	10^−6^/10^2^	10^−5^/10^−3^	10^−6^/10^−3^	10^−6^/10^2^	10^−6^/10^2^	10^−6^/10^−3^
Bipolar disorder	10^−6^/10^−3^	10^−5^/10^−3^	10^−5^/10^−3^	10^−6^/10^−3^	10^−6^/10^−3^	10^−6^/10^−3^
Schizophrenia	10^−5^/10^−3^	10^−5^/10^−3^	10^−5^/10^−3^	10^−5^/10^−3^	10^−5^/10^−3^	10^−6^/10^−3^

## Data Availability

The original data supporting the conclusions of this article is openly available in Zenodo at 10.5281/zenodo.13514193.

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
