# Peer review of "Classifying Unconscious, Psychedelic, and Neuropsychiatric Brain States with Functional Connectivity, Graph Theory, and Cortical Gradient Analysis"

_brainsci, 2024, doi:10.3390/brainsci14090880_

Round 1

Reviewer 1 Report

Comments and Suggestions for Authors

The authors propose an interesting method for classifying unconscious, psychedelic, and neuropsychiatric brain states. However, it is necessary to address the following comments

-        Improve the introduction or add a section of related work where MRI and machine learning based work is mentioned (what they did and what they got).

-        In the last paragraph of the introduction, summarize the main contributions of your paper.

-        Improve image quality

-        The datasets were taken with different MRI equipment, which can create a bias in the information, how do you deal with this point?

-        The authors mention the datasets used in the development of the work, sometimes it is a bit confusing, for example in some cases they mention that T1 and T2 images were obtained, which ones were used? At the end of the explanation of the datasets, it would be advisable to provide a detailed summary table of the information used for each data set.

-        Add a table showing the results of hyperparameter tuning.

-        Line 190 mentions the age and gender of the participants in a dataset, which are 12 males and 14 females, ranging in age from 27 to 64. How might this gender and age variation affect the results? Isn't there an information bias?

-        Compare your results with those found in the literature, a suggestion might be a comparison table.

-        What is the direction of the future work according to the limitations of the present?

-        Add a Conclusions Section.

Author Response

Comments 1: Improve the introduction or add a section of related work where MRI and machine learning based work is mentioned (what they did and what they got).

Response 1: Thank you for your comment. We revised the introduction and added one paragraph reviewing the fMRI-based machine learning studies, especially related to support vector machine.

“Notably, SVM has proven to be a robust and versatile method for brain pattern recognition, as extensively reviewed in the literature [31–34]. For instance, SVM has successfully classified major depression based on fMRI signal differences observed dur-ing the emotional processing of sad faces [35]. Additionally, SVM has demonstrated over 90% accuracy in distinguishing the progression of Alzheimer’s disease by analyz-ing brain networks reconstructed from resting-state fMRI data [36]. Moreover, SVM has shown promising results in the classification of schizophrenia and bipolar disorder, further highlighting its utility in neuropsychiatric research [37].” (Line 58)

  1. Costafreda, S.G.; Fu, C.H.; Picchioni, M.; Toulopoulou, T.; McDonald, C.; Kravariti, E.; Walshe, M.; Prata, D.; Murray, R.M.; McGuire, P.K. Pattern of Neural Responses to Verbal Fluency Shows Diagnostic Specificity for Schizophrenia and Bipolar Disorder. BMC Psychiatry 2011, 11, 18, doi:10.1186/1471-244X-11-18.
  2. Jr, L.S. Application of Support Vector Machine on fMRI Data as Biomarkers in Schizophrenia Diagnosis: A Systematic Review. Frontiers in Psychiatry 2020, 11.
  3. Orrù, G.; Pettersson-Yeo, W.; Marquand, A.F.; Sartori, G.; Mechelli, A. Using Support Vector Machine to Identify Imaging Biomarkers of Neurological and Psychiatric Disease: A Critical Review. Neuroscience & Biobehavioral Reviews 2012, 36, 1140–1152, doi:10.1016/j.neubiorev.2012.01.004.
  4. Bondi, E.; Maggioni, E.; Brambilla, P.; Delvecchio, G. A Systematic Review on the Potential Use of Machine Learning to Classify Major Depressive Disorder from Healthy Controls Using Resting State fMRI Measures. Neuroscience & Biobehavioral Reviews 2023, 144, 104972, doi:10.1016/j.neubiorev.2022.104972.
  5. Fu, C.H.Y.; Mourao-Miranda, J.; Costafreda, S.G.; Khanna, A.; Marquand, A.F.; Williams, S.C.R.; Brammer, M.J. Pattern Classification of Sad Facial Processing: Toward the Development of Neurobiological Markers in Depression. Biological Psy-chiatry 2008, 63, 656–662, doi:10.1016/j.biopsych.2007.08.020.
  6. Khazaee, A.; Ebrahimzadeh, A.; Babajani-Feremi, A.; Alzheimer’s Disease Neuroimaging Initiative Classification of Patients with MCI and AD from Healthy Controls Using Directed Graph Measures of Resting-State fMRI. Behav Brain Res 2017, 322, 339–350, doi:10.1016/j.bbr.2016.06.043.
  7. Rashid, B.; Arbabshirani, M.R.; Damaraju, E.; Cetin, M.S.; Miller, R.; Pearlson, G.D.; Calhoun, V.D. Classification of Schiz-ophrenia and Bipolar Patients Using Static and Dynamic Resting-State fMRI Brain Connectivity. NeuroImage 2016, 134, 645–657, doi:10.1016/j.neuroimage.2016.04.051.

Comments 2: In the last paragraph of the introduction, summarize the main contributions of your paper.

Response 2: Thank you for your suggestion. We have added a sentence in the last paragraph of the introduction to succinctly summarize the main contributions of our paper.

“To this end, we present a novel brain state classification methodology that integrates functional connectivity, graph-theoretical metrics, and cortical gradients, demonstrating robust performance across unconscious, psychedelic, and neuropsychiatric states with enhanced generalizability.” (Line 116)

Comments 3: Improve image quality

Response 3: We attached the high-quality (450 PPI) figure image files along with the revised manuscript.

Comments 4: The datasets were taken with different MRI equipment, which can create a bias in the information, how do you deal with this point?

Response 4: We appreciate your concern regarding the potential bias introduced by using different MRI equipment across datasets. We agree that this could introduce unwanted variability. However, in Figures 2 and 3, we trained and validated our models within the same dataset, where all data were collected using the same MRI equipment.

For the cross-dataset evaluation in Figure 4, we applied our models to a different dataset that was acquired using different MRI equipment. Importantly, this new dataset contained its own baseline, which helped mitigate the issue of scanner-specific bias. Despite the variation in equipment, our results showed that the models could still generalize to some degree, successfully differentiating between baseline and atypical brain states. This indicates that our approach is robust and captures neural differences, rather than being overly influenced by the characteristics of the MRI scanners used.

Comments 5: The authors mention the datasets used in the development of the work, sometimes it is a bit confusing, for example in some cases they mention that T1 and T2 images were obtained, which ones were used? At the end of the explanation of the datasets, it would be advisable to provide a detailed summary table of the information used for each data set.

Response 5: We apologize for the confusion regarding the description of the datasets and the imaging modalities used. To clarify, T1 anatomical images were only utilized for co-registration during the data pre-processing stage. On the other hand, T2*-weighted images were used for fMRI to capture the blood-oxygen-level-dependent (BOLD) signal.

To enhance clarity and avoid any further confusion, we have streamlined the explanation of the anatomical imaging process in the manuscript. Additionally, we have included a summary table (Table 1, Line 149).

Comments 6: Add a table showing the results of hyperparameter tuning.

Response 6: Thank you for the suggestion. We added the table showing the hyperparameter tuning results in Table 2 (Line 477)

Comments 7: Line 190 mentions the age and gender of the participants in a dataset, which are 12 males and 14 females, ranging in age from 27 to 64. How might this gender and age variation affect the results? Isn't there an information bias?

Response 7: Thank you for highlighting the potential impact of gender and age variation on our results. We acknowledge that these factors can introduce bias. However, for most of our datasets, particularly those involving non-neuropsychiatric conditions, we used a within-subject design, which inherently controls individual differences by comparing brain states within the same participants. This approach minimizes the influence of demographic variables like age and gender. Thus, while demographic variability is an important consideration, our methodology has been designed to mitigate its potential impact on the results. We emphasized this point in the revised manuscript.

“Except for neuropsychiatric conditions, we employed a within-subject design, by comparing two brain states (baseline vs. atypical) within the same participants.” (Line 146)

Comments 8: Compare your results with those found in the literature, a suggestion might be a comparison table.

Response 8: Thank you for your suggestion. We have incorporated a comprehensive comparison within the discussion section. In the newly added paragraph, we compare the performance of our models with previous research findings, detailing how our results align with or differ from existing studies.

“The performance of our models aligns with previous research findings. For instance, earlier studies on sleep stage detection using functional connectivity data re-ported approximately 80% accuracy, which is comparable to our results [30]. In the context of anesthesia, previous machine learning analyses using functional connectivity achieved an AUC of 0.67 [22], whereas our analysis significantly outperformed this, approaching nearly 100% AUC. For ADHD and bipolar disorder, typical studies report accuracy ranges from 60% to 80%, and our model performed within this expected range [74,75]. Regarding schizophrenia, a functional connectivity-based study using the same dataset as ours reported 70% accuracy [76], while a machine learning approach using graph theoretical data with the same dataset achieved 79% accuracy [77]. Our performance, with a balanced accuracy of 72%, is consistent with these findings. Moreover, it is important to highlight that our methodology has a distinct advantage in terms of generalizability, making it applicable across various brain states.” (Line 664)

  1. Campbell, J.M.; Huang, Z.; Zhang, J.; Wu, X.; Qin, P.; Northoff, G.; Mashour, G.A.; Hudetz, A.G. Pharmacologically Informed Machine Learning Approach for Identifying Pathological States of Unconsciousness via Resting-State fMRI. NeuroImage 2020, 206, 116316, doi:10.1016/j.neuroimage.2019.116316.
  2. Tagliazucchi, E.; von Wegner, F.; Morzelewski, A.; Borisov, S.; Jahnke, K.; Laufs, H. Automatic Sleep Staging Using fMRI Functional Connectivity Data. NeuroImage 2012, 63, 63–72, doi:10.1016/j.neuroimage.2012.06.036.
  3. Taspinar, G.; Ozkurt, N. A Review of ADHD Detection Studies with Machine Learning Methods Using rsfMRI Data. NMR in Biomedicine 2024, 37, e5138, doi:10.1002/nbm.5138.
  4. Campos-Ugaz, W.A.; Palacios Garay, J.P.; Rivera-Lozada, O.; Alarcón Diaz, M.A.; Fuster-Guillén, D.; Tejada Arana, A.A. An Overview of Bipolar Disorder Diagnosis Using Machine Learning Approaches: Clinical Oppor-tunities and Challenges. Iran J Psychiatry 2023, 18, 237–247, doi:10.18502/ijps.v18i2.12372.
  5. Kim, M.; Seo, J.W.; Yun, S.; Kim, M. Bidirectional Connectivity Alterations in Schizophrenia: A Multivariate, Machine-Learning Approach. Front. Psychiatry 2023, 14, doi:10.3389/fpsyt.2023.1232015.
  6. Tavakoli, H.; Rostami, R.; Shalbaf, R.; Nazem-Zadeh, M.-R. Diagnosis of Schizophrenia and Its Subtypes Using MRI and Machine Learning 2024, 2024.08.09.24311726.

Comments 9: What is the direction of the future work according to the limitations of the present?

Response 9: As mentioned in the manuscript (Line 742), future work should focus on several critical areas to enhance the robustness and applicability of our findings. First, validating with separate datasets with higher number of samples across all conditions will be essential to verify the stability and reliability of our results. Second, expanding the analysis to include node-level features, alongside network-level metrics, could provide more detailed insights into the specific brain regions driving the classifications. Additionally, addressing potential scanner effects by harmonizing data collected from different MRI scanners will be crucial to reducing variability and improving the generalizability of our methods. These steps will guide future research in refining our approach and expanding its utility.

Comments 10: Add a Conclusions Section.

Response 10: Thank you for the suggestion. We modified the structure of the previous Discussion section and separated the Conclusions section.

Reviewer 2 Report

Comments and Suggestions for Authors

Review for manuscript brainsci-3176361-v1 “Classifying unconscious, psychedelic, and neuropsychiatric brain states with functional connectivity, graph theory, and cortical gradient analysis” by H. Jang et al.

            This manuscript reports a study investigating the efficacy of a support vector machine (SVM) learning model for discriminating among several examples of three categories of global resting brain states (unconsciousness, hallucinogenic psychedelic states, neuropsychiatric disorders). The SVM model utilized by the authors integrated three sets of features extracted from fMRI data: functional connectivity patterns, graph-theoretic metrics, and cortical gradients. The SVM model incorporated nested cross-validation, and feature ensembles were created according to three different schemes: full feature integration (all features combined into a single ensemble), hard voting (base feature sets remained separate, each providing a single prediction, with the majority prediction determining a classification), and soft voting (prediction scores from each base feature set were summed to produce a final score for each final category). The authors found that, in general, the SVM model utilizing a soft voting scheme performed the best out of all the models. However, different models exhibited varying degrees of generalizability across different datasets depending on brain states and feature sets. Moreover, an analysis of the relative importance of different features for the performance of the models showed that successful model discrimination of different global brain states depended on different features. The authors interpret this as indicating that the neural mechanisms underlying these brain states are variable and thus a tailored approach to feature selection is necessary for the accurate classification of specific brain states.

            I believe this study may provide a very useful and novel machine learning methodology for studying the differences between global brain states. The study also catalogs some interesting observations of several global brain states using this method. In general, I find the methods and analysis to be well-reported and sufficiently rigorous. My main concern with this study is the absence of a wakeful resting brain state as a baseline comparison (see questions below). Moreover, I have a few clarification questions that, if addressed, will improve the readability and ease of understanding of the manuscript. Some of the latter questions arise from the fact that I only have a basic understanding of graph theory and cortical gradient analysis techniques. However, I do think it is important to clarify such questions to facilitate the understanding of this study for a general readership of this paper.

My questions and concerns are as follows:

Lines 111 - 299, Section 2.1.1 Experimental Conditions: The authors utilized multiple different datasets indexing specific examples of three global brain state categories. The basic details of the data recording and experimental methodology are well-reported as far as I can tell. However, I question the use of data obtained from different subjects under differing methods; could not the idiosyncrasies in these factors contribute to the discriminability of the different models aside from differences in the neural mechanisms underlying the different brain states? For example, could the models discriminate brain state A versus B simply because these are two entirely different groups of subjects, each with their own idiosyncratic brain state patterns. It seems to me that this situation might be like that found when comparing mean brain activation between groups; one does not know if differences reflect true neural differences or some other idiosyncratic factor. To address this, one typically contrasts two conditions (usually an activation condition and baseline condition) within each subject to remove signals related to idiosyncratic biophysical and neurogeometrical factors and then compare these difference values between the groups. It does not seem that the datasets used by this study allow for this, unless I am misunderstanding the details of the data collection.

More generally, why did the authors not include wakeful states in their analysis too allow a “baseline” to compare the other states with? This might reduce some of the ambiguity when interpreting whether two (or more) separate brain states have similar or different underlying neural mechanisms (see related comment further below).

Lines 332- 333: The authors state that they “calculated the mean and standard deviation of functional connectivity values from the 400 × 400 matrix”. There are many ways to quantify functional connectivity. How was the functional connectivity specifically quantified here? By a linear correlation of the average time course for each predefined cortical area? Some other metric?

Lines 341 – 343: The authors state that “[e]ight unweighted brain graphs (whole brain and seven subnetworks) were constructed after binarizing the functional connectivity matrices at 90% sparsity, preserving the top 10% of edges”. It is difficult to understand what the authors mean by “top 10% of edges” here without knowing the specific statistic used to quantify functional connectivity (see comment immediately above). For example, if this statistic is a linear correlation that could be positive or negative, then does “top 10%” mean “above the 90th percentile”, which are only positive correlations? Why ignore negative correlations? On the other hand, if the absolute value of the correlation is what is relevant, then the “top 10%” may include positive and negative correlations of sufficient magnitude. Please clarify.

Lines 370 – 372: A similar question about the meaning of “top 10% of weighted connections” arises here, if the connection weights can be positive or negative. Please clarify.

Lines 372 – 373: Was the “gradient” indexed in the gradient analysis a gradient of connectivity profiles between cortical areas? That is, areas with similar profiles are located near each other and areas with dissimilar profiles are located far away from each other on the gradient map? Please clarify.

Lines 512 – 518: The authors state that “[i]t is crucial to note that these cross-dataset evaluations do not indicate the models’ ability to differentiate between the train and test conditions directly. Rather, they reflect the capacity to distinguish between normal wakefulness and altered brain states. For instance, if a model trained on N2 sleep data achieves high accuracy when tested on LSD data, this does not mean it is differentiating between sleep and LSD states. Instead, it suggests the model is effectively identifying features that distinguish normal wakefulness from altered brain states, regardless of the specific alteration”. This is a remarkable statement, if true. However, it is unclear to me how the authors can conclude this when a global brain state of normal wakefulness was not included as one of the analyzed datasets or used as a contrast against the other brain states (see my earlier comment on this issue above). Are the authors arguing that the N2 sleep state has some similarities to the waking state, as does the LSD psychedelic state, and it is in virtue of these similarities that the N2 sleep state-trained model can then discriminate the LSD state? If so, would this not be contradicted by their suggestion on lines 665 – 669 of the present manuscript that these two states have an inverse relationship? Could it be possible that there is some other neural similarity between the two states that does not reflect the waking state? Please clarify.

To conclude, in terms of reporting a “methods” study I think this manuscript is quite good (subject to clarification of the technical issues I raise above). However, it terms of a “discovery” study reporting observed differences between brain states, I believe the current manuscript is limited due to the baseline issues I raise above.

Comments on the Quality of English Language

The English language is satisfactory and only requires minor editing.

Author Response

Comments 1: Lines 111 - 299, Section 2.1.1 Experimental Conditions: The authors utilized multiple different datasets indexing specific examples of three global brain state categories. The basic details of the data recording and experimental methodology are well-reported as far as I can tell. However, I question the use of data obtained from different subjects under differing methods; could not the idiosyncrasies in these factors contribute to the discriminability of the different models aside from differences in the neural mechanisms underlying the different brain states? For example, could the models discriminate brain state A versus B simply because these are two entirely different groups of subjects, each with their own idiosyncratic brain state patterns. It seems to me that this situation might be like that found when comparing mean brain activation between groups; one does not know if differences reflect true neural differences or some other idiosyncratic factor. To address this, one typically contrasts two conditions (usually an activation condition and baseline condition) within each subject to remove signals related to idiosyncratic biophysical and neurogeometrical factors and then compare these difference values between the groups. It does not seem that the datasets used by this study allow for this, unless I am misunderstanding the details of the data collection.

More generally, why did the authors not include wakeful states in their analysis too allow a “baseline” to compare the other states with? This might reduce some of the ambiguity when interpreting whether two (or more) separate brain states have similar or different underlying neural mechanisms (see related comment further below).

Response 1: We apologize for the lack of clarity regarding baseline conditions. Baseline conditions were indeed incorporated into all analyses.

In Figures 2 and 3, each model was trained and tested on the same dataset, enabling direct comparison of baseline and atypical states within the same subject group, thereby minimizing scanner-related confounds.

In Figure 4, we extended our analysis by applying the trained models to an entirely different dataset, which includes both baseline and atypical conditions. Importantly, this test dataset had its own baseline, allowing us to assess the model performance in distinguishing between baseline and atypical brain states within a new population. Our results indicate that the models retain generalizability, successfully differentiating these states even in a dataset that the models had not previously encountered. This suggests that our methodology captures the underlying neural mechanisms of the brain states of interest rather than merely reflecting dataset-specific idiosyncrasies. We explicitly mentioned this in the result section to prevent confusion. We also edited the text in Figures 2 and 3 to clarify the baseline and atypical conditions within each dataset.

“Importantly, this test dataset had its own baseline, allowing us to assess the model performance in distinguishing between baseline and atypical brain states within a new population.” (Line 529)

Comments 2: Lines 332- 333: The authors state that they “calculated the mean and standard deviation of functional connectivity values from the 400 × 400 matrix”. There are many ways to quantify functional connectivity. How was the functional connectivity specifically quantified here? By a linear correlation of the average time course for each predefined cortical area? Some other metric?

Response 2: We apologize for lack of clarity. We calculated Pearson correlation coefficient between the time courses for a given pair of ROIs. We updated the method section to address this.

“Functional connectivity matrix (400 × 400) was calculated by Pearson correlation of the time courses. We then calculated the mean and standard deviation of functional connectivity values.” (Line 342)

Comments 3: Lines 341 – 343: The authors state that “[e]ight unweighted brain graphs (whole brain and seven subnetworks) were constructed after binarizing the functional connectivity matrices at 90% sparsity, preserving the top 10% of edges”. It is difficult to understand what the authors mean by “top 10% of edges” here without knowing the specific statistic used to quantify functional connectivity (see comment immediately above). For example, if this statistic is a linear correlation that could be positive or negative, then does “top 10%” mean “above the 90th percentile”, which are only positive correlations? Why ignore negative correlations? On the other hand, if the absolute value of the correlation is what is relevant, then the “top 10%” may include positive and negative correlations of sufficient magnitude. Please clarify.

Lines 370 – 372: A similar question about the meaning of “top 10% of weighted connections” arises here, if the connection weights can be positive or negative. Please clarify.

Response 3: The reviewer is correct that we first used Pearson correlation between the time courses for a given pair of ROIs to generate the functional connectivity matrix. The Pearson correlation coefficients in this matrix were then sorted, and the top 10% of the highest coefficients were selected.  This "top 10%" indeed corresponds to values above the 90th percentile, retaining only positive correlations.

We appreciate the reviewer highlighting the importance of negative correlations in fMRI functional connectivity analysis. While negative correlations can provide valuable information (Parente et al., 2018), their interpretation remains a subject of ongoing debate (Zhan et al., 2017; Schwarz and McGonigle, 2011). In our analysis, we focused on the top 10% of connectivity values, which is a common practice in graph theory analysis of functional connectivity (Timmermann et al., 2023; Hallquist and Hillary, 2018), rather than explicitly excluding negative correlations.

In the revised manuscript, we have added the following sentence to clarify this point:

“Eight unweighted brain graphs (whole brain and seven subnetworks) were con-structed after binarizing the functional connectivity matrices at 90% sparsity (10% edge density), preserving the top 10% of edges, which is consistent with the common prac-tice in the literature [62,63]” (Line 351)

  1. Hallquist, M.N.; Hillary, F.G. Graph Theory Approaches to Functional Network Organization in Brain Disorders: A Critique for a Brave New Small-World. Network Neuroscience 2018, 3, 1–26, doi:10.1162/netn_a_00054.
  2. Timmermann, C.; Roseman, L.; Haridas, S.; Rosas, F.E.; Luan, L.; Kettner, H.; Martell, J.; Erritzoe, D.; Tagliazuc-chi, E.; Pallavicini, C.; et al. Human Brain Effects of DMT Assessed via EEG-fMRI. Proceedings of the National Academy of Sciences 2023, 120, e2218949120, doi:10.1073/pnas.2218949120.

Comments 5: Lines 372 – 373: Was the “gradient” indexed in the gradient analysis a gradient of connectivity profiles between cortical areas? That is, areas with similar profiles are located near each other and areas with dissimilar profiles are located far away from each other on the gradient map? Please clarify.

Response 5: Thank you for your question. Yes, the gradient indexed in our gradient analysis reflects the similarity of connectivity profiles between cortical areas. Specifically, cortical areas with similar functional connectivity profiles are positioned near each other on the gradient map, while areas with more dissimilar connectivity profiles are located farther apart. This approach captures the continuous variation in connectivity patterns across the cortex, allowing us to reveal the hierarchical organization of brain networks. We have clarified this point in the manuscript to ensure the concept is more transparent to readers.

“In result, cortical areas with similar functional connectivity profiles are positioned near each other on the gradient map, while areas with more dissimilar connectivity profiles are located farther apart.” (Line 388)

Comments 7: Lines 512 – 518: The authors state that “[i]t is crucial to note that these cross-dataset evaluations do not indicate the models’ ability to differentiate between the train and test conditions directly. Rather, they reflect the capacity to distinguish between normal wakefulness and altered brain states. For instance, if a model trained on N2 sleep data achieves high accuracy when tested on LSD data, this does not mean it is differentiating between sleep and LSD states. Instead, it suggests the model is effectively identifying features that distinguish normal wakefulness from altered brain states, regardless of the specific alteration”. This is a remarkable statement, if true. However, it is unclear to me how the authors can conclude this when a global brain state of normal wakefulness was not included as one of the analyzed datasets or used as a contrast against the other brain states (see my earlier comment on this issue above). Are the authors arguing that the N2 sleep state has some similarities to the waking state, as does the LSD psychedelic state, and it is in virtue of these similarities that the N2 sleep state-trained model can then discriminate the LSD state? If so, would this not be contradicted by their suggestion on lines 665 – 669 of the present manuscript that these two states have an inverse relationship? Could it be possible that there is some other neural similarity between the two states that does not reflect the waking state? Please clarify.

Response 7: Thank you for raising this important point. We acknowledge that our original statement may have been unclear, and we appreciate the opportunity to clarify our intentions.

In the cited paragraph, our aim was to emphasize that when a model is trained on one altered brain state (e.g., N2 sleep) versus its baseline (normal wakefulness) and then tested on another altered state (e.g., LSD) versus its own baseline (e.g., before LSD infusion), we do not expect the model to differentiate between the two specific altered states (sleep vs. LSD). Instead, the model's primary function is to distinguish between altered states (e.g., LSD) and normal wakefulness, based on the features it learned during training on the specific altered state (e.g., sleep-related features for the N2 sleep model).

To illustrate, if a model trained on N2 sleep data yields an AUC lower than 0.5 when tested on LSD data, this doesn't contradict our findings. To clarify further, if we suggest that N2 sleep and LSD are inversely related, the N2 sleep model would learn a boundary distinguishing baseline (normal wakefulness) from N2 sleep. If the LSD state resides at the opposite of the N2 sleep state along the eigenvector that separates normal wakefulness from N2 sleep, then in a test scenario for LSD vs. baseline, the baseline state of LSD might appear "relatively more N2 sleep-like" compared to the LSD state, leading to an AUC lower than 0.5. (It's important to remember that the test dataset has its own baseline and atypical data points.)

We have revised the paragraph in the manuscript to better articulate this concept:

 “It's important to emphasize that these cross-dataset evaluations do not measure the model’s ability to differentiate between the two atypical brain states in the train and test datasets. Instead, they assess the model's capacity to classify the test dataset into two groups based on the features it learned during training. For example, if a model trained on N2 sleep data shows high accuracy when tested on general anesthesia data, it doesn't mean it's directly differentiating between sleep and anesthesia. Rather, it suggests that the features the model learned to distinguish normal wakefulness from sleep are also somewhat effective in distinguishing normal wakefulness from general anesthesia.” (Line 532)

Round 2

Reviewer 1 Report

Comments and Suggestions for Authors

Thank you for considering the comments, but it is necessary to improve the conclusions by mentioning the main contributions and adding numerical results.

Reviewer 2 Report

Comments and Suggestions for Authors

I thank the authors for their clarifications in response to my questions. I am satisfied with the revised version of the manuscript.

Author Response

Thank you for your help